# Influence of Land Use on the C and N Status of a C_4_ Invasive Grass in a Semi-Arid Region: Implications for Biomonitoring

**DOI:** 10.3390/plants10050942

**Published:** 2021-05-09

**Authors:** Edison A. Díaz-Álvarez, Erick de la Barrera

**Affiliations:** 1Instituto de Investigaciones Forestales, Universidad Veracruzana, Parque Ecológico “El Haya”, Carretera Antigua a Coatepec-Coapexpan, Xalapa, Veracruz 91070, Mexico; edisondiaz@uv.mx; 2Instituto de Investigaciones en Ecosistemas y Sustentabilidad, Universidad Nacional Autónoma de México, Antigua Carretera a Pátzcuaro 8701, Col. Ex-Hacienda de San José del Cerrito, Morelia Michoacán 58190, Mexico

**Keywords:** biomonitoring, carbon emissions, environmental pollution, global ecology, nitrogen deposition, stable isotopes

## Abstract

Biomonitoring of atmospheric pollution is an increasingly accepted practice. However, most existing biomonitors are usually epiphytic species from mesic environments. This work assessed the suitability of buffelgrass (*Cenchrus ciliaris*), an invasive C_4_ grass in northwestern Mexico, as a biomonitor, by means of the spatial distribution of the carbon and nitrogen content and isotopic signatures for grass samples collected from urban, agricultural, and natural areas throughout the state of Sonora. We found the highest tissue carbon content of 45.6% (on a dry weight basis) and highest nitrogen content of 3.31% for buffelgrass from the Yaqui Valley. We also found the lowest δ^13^C of −15.9‰, and the highest δ^15^N of 16.7‰ in the same region. In contrast, the lowest carbon and nitrogen content of 39.4 and 1.49% were found for Bahía de Kino and Río Sonora mountains, respectively. The lowest δ^15^N of 2.18‰ and the highest δ^13^C of −13.7‰ were measured for two remote locations. These results show the influence that pollutant emissions, including agriculture and transportation, have on elemental and isotopic composition of vegetation. Buffelgrass is most adequate for tracking carbon and nitrogen emissions in arid environments and for determining alterations on nitrogen soil reactions, as a first approximation for saturation.

## 1. Introduction

Anthropic activities, mainly those related to urbanization and agriculture, cause the release and subsequent deposition of substantial amounts of pollutants. Carbon and nitrogen compounds are two groups of pollutants of special concern given their noxious environmental effects, the better-known of which are climate change and nitrogen deposition [1,2,3]. Atmospheric pollution not only has significant impacts on natural environments, since it is considered one of the leading drivers of biodiversity loss, but it also is a matter of human health, as air pollution causes premature mortality and various diseases that have become a global burden [4,5,6].

Various methods can be used for monitoring atmospheric pollution, with the gold standard being the deployment of automated monitoring stations that record the atmospheric concentrations of various atmospheric pollutants such as ozone, sulfur dioxide, and heavy metals in a continuous and reliable manner [7]. However, because multiple point measurements are required to monitor vast areas, the deployment of numerous monitoring stations is necessary, which can represent excessive costs of operation, maintenance, and space. A more economical alternative is the use of passive collectors, such as those that are commonly used to quantify nitrogen deposition. This methodology is effective in tracking atmospheric pollutants in large areas, but has the disadvantage of requiring that the collectors be replaced every few weeks and that the measurements are sensitive to variations of temperature, air movement, and humidity [8,9].

Another alternative is the use of biomonitors, i.e., plants that occur naturally in regions of interest, which have been shown to be particularly suited for assessing environmental pollution by carbon and nitrogen emissions [10,11]. Biomonitors further allow the integration of different environmental factors, including water availability, air temperature, and atmospheric pollution, as well as the respective changes in such parameters in both natural and modified environments [8,12,13]. Of particular promise are the so-called atmospheric biomonitors, plants whose mineral nutrition depends mostly on deposited or gaseous atmospheric sources. Alas, such epiphytic species tend to occur predominantly in mesic environments and tend to be less abundant in arid and semi-arid regions, thus limiting the widespread use of this “biotechnology”. To address this limitation, the elemental and isotopic composition of grass biomonitors have recently been shown to detect modifications to the nitrogen cycle in natural and modified environments, including within arid regions [8,12,13,14,15]. Stable isotopes have been deemed as a powerful analytical tool that allows an integral assessment of various biogeochemical and ecological processes [16]. In particular, as the carbon and nitrogen move through different planetary compartments, they are subject to different reactions and their consequent fractionations. This also applies to pollutants emitted by different human activities. For example, while −8‰ is a typical δ^13^C value for the air in natural environments, the carbon emissions from coal, gasoline, diesel, and natural gas combustion are substantially more negative, ranging from −42 to −25‰, a signal that is picked up by photosynthetic biomonitors [11,17,18,19,20,21]. For nitrogen emissions, oxidized compounds emitted by vehicular and industrial activities tend to have positive δ^15^N values, contrasting with the very negative values measured for reduced compounds such as those emitted by agriculture and animal husbandry [8,10,15]. 

We assessed the potential of *Cenchrus ciliaris* (common name is buffelgrass) as a candidate for biomonitoring carbon and nitrogen pollution in a semi-arid region from northwestern Mexico, where this plant is abundant, and the environmental conditions disallow the widespread occurrence of epiphytes. Indeed, with annual precipitations as low as 140 mm, this plant is most able to germinate, develop, and spread effectively throughout the study region, reasons for which it was introduced to the southern United States in the 1940s and to northwestern Mexico in the 1950s for forage and fodder production [22,23,24,25]. Buffelgrass has since escaped and invaded vast areas, displacing numerous local species [26,27,28]. In fact, this African grass has become one of the most important invasive plants worldwide, as a result of its phenological and ecophysiological characteristics, including rapid growth and tolerance to drought, which stems from its C_4_ photosynthesis [29].

Even though the human population in the state of Sonora amounts to 2 million and that productive activities presumably represent the emission of large amounts of atmospheric pollutants, such that their monitoring is required by the Mexican environmental regulation, automated monitoring stations have been deployed in Hermosillo, the state capital, and Ciudad Obregón, but they are not operational [30,31,32]. Monitoring air quality in the rest of the state is irregular, at best, and limited to sporadic measurements of SO_2_ and particulate matter [32]. For these reasons, taking advantage of the regional abundance of this invasive grass, we measured its carbon and nitrogen elemental and isotopic composition by means of sampling leaves and seeds of this plant from 34 sites with various land uses in Sonora, Mexico (Figure 1).

## 2. Results

### 2.1. Environmental Characteristics 

The mean monthly temperature averaged 23.5 ± 0.15 °C among the 34 collecting sites from Sonora, Mexico, with a minimum temperature of the coldest month of 7.5 ± 0.25 °C and a maximum temperature of the warmest month of 38.0 ± 0.15 °C (Table 1). The annual precipitation was 287 ± 20 mm throughout the study region. The soil at the study sites was mostly alluvial in origin, with a sandy clay loam texture and an electrical conductivity of 0.71 ± 0.14 mmhos/cm. The primary soil type for three quarters of the sites was either a cambisol, a luvisol, or a vertisol. Although a considerable variation was found in the study region, a Principal Components Analysis revealed that the environment was essentially indistinguishable among the various land-use classes considered (Figure 2; Table 2).

### 2.2. Elemental and Isotopic Composition for Buffelgrass

Over the study region the leaf carbon content of buffelgrass ranged from 39.4% (dry weight basis), for plants from site No. 21 located near the coast, to 45.6%, for plants from site No. 10 within the Yaqui Valley, averaging 42.1 ± 0.16% throughout the study region (Figure 3A). The lowest seed carbon content of 42.8% was also measured for the plants collected at site No. 21, while the highest seed carbon content of 51.4% was found for site no.31 in a natural protected area (Figure 3B). For medium urban sites, carbon content of leaves was higher than for rural and small urban sites. The carbon content for seeds averaged 47.5 ± 0.89% and it was significantly higher than for the leaves (*p* = 0.001; Table 3 and Table A1; Figure 4). Land use had a significant effect on the carbon content of *C. ciliaris* as significant differences were found among the five land uses (*p* = 0.001). Seeds from small urban sites had a lower carbon content than those from rural and medium urban sites. The interaction organ × land use was also significant (*p* = 0.002; Table 3; Figure 4A).

The lowest leaf nitrogen content was 1.49%, found for the plants collected at site no.16, a remote rural location close to the Río Sonora mountains. Conversely, the highest nitrogen content of 3.31% was found for plants growing at site no.14, also a rural location but, in this case, close to a secondary road and an agricultural field (Figure 3C). On average, the leaf N content reached 2.52 ± 0.05% and it was significantly lower, 1.60 ± 0.04%, than for the seeds (*p* = 0.001; Table 3). The leaves from agricultural sites had the highest nitrogen content, small urban sites had the lowest. The lowest nitrogen content for seeds was found for plants from site no. 12 that reached 0.99%, while the highest content was 2.35% for site no. 11 in a rural area (Figure 3D). Land use had a significant effect on the nitrogen content of *C. ciliaris* (*p* = 0.002). For example, this parameter was higher for seeds from medium urban sites than for those from rural sites. However, the interaction between organ × land use was not significant (*p* = 0.057; Table 3; Figure 4A).

The C:N ratio averaged 17.98 ± 0.49 for leaves, ranging from 11.9 at site no. 29 to 28.9 at site no. 16 (Figure 3E). Significant differences were observed among land uses (*p* = 0.002; Table 3). C:N ratio for leaves from large urban was higher than for agriculture. For the seeds, the C:N ratio averaged 31.85 ± 0.96, with the highest value of 49.6 found for the plants from site no. 31, and the lowest value in site no.29, which reached 20.5 (Figure 3F). Significant differences were also observed for the C:N ratio between leaves and seeds (*p* = 0.001; Table 3). In addition, the C:N ratios among land uses were different (*p* = 0.001), and the interaction between organ × land use for *C. ciliaris* was significant (*p* = 0.015; Table 3). Seeds collected from rural sites had the highest C:N ratios, while the lowest were from agricultural sites.

The δ^13^C values for the leaves ranged from –15.9‰ for plants growing in site no. 17, in the city of Hermosillo to −13.7‰ for plants collected in site no. 29, near a road and a wheat field, the land use had a significant effect on this parameter (*p* = 0.002; Table 3; Figure 5A). The δ^13^C values for the leaves from small urban sites and the lowest was found in large urban sites (Figure 4). The average δ^13^C value for seeds was −14.3 ± 0.30‰. The lowest δ^13^C value for seeds reached −17.2‰ for plants growing at site no. 31 in a natural protected area, while the highest value of −13.9‰ was measured for plants growing at site no. 13 in a small urban center, significant differences were observed among land uses (*p* = 0.002; Table 3; Figure 5B). The δ^13^C values for *C. ciliaris* were significant for the organs (*p* = 0.001), also, the interaction between organ × land use was significant (*p* = 0.003; Figure 4B; Table 3). In addition, the seeds from small urban sites were more enriched than those from rural sites (Figure 4).

The δ^15^N values for leaves ranged from 2.18 to 16.7‰, respectively for plants collected near a highway in an isolated area (site no. 12) and a site where urban and cultivated fields join (site no. 9). Significant differences were found among the different classes of land use (*p* = 0.001; Table 3; Figure 4B and Figure 5C). The δ^15^N values for the leaves of buffelgrass averaged 8.37 ± 0.20‰ and 8.77 ± 0.26‰ for the seeds, throughout the study area. The lowest δ^15^N value was found for leaves from small urban sites, in contrast agriculture and rural sites had the highest δ^15^N value (Figure 4). The lowest value of 2.71‰ was found for seeds at the most remote point in site no. 12, the highest of 16.4‰ was found near a highway in site no. 10 (Figure 4B and Figure 5D). No differences were found for the δ^15^N values between leaves and seeds (*p* = 0.416; Table 3). On the contrary, the interaction between organ × land use was significant (*p* = 0.012; Table 3). For seeds, the δ^15^N values were highest for medium urban and lowest for large urban sites (Figure 4).

## 3. Discussion

Plant carbon content is influenced by multiple environmental factors, including water and nutrient availability, the prevailing CO_2_ concentration where plants grow [33]. Buffelgrass can become very successful in allocating biomass to its organs under adverse environmental conditions, owing to physiological traits such as the use of C_4_ photosynthesis, which improves their water use efficiency [34]. Indeed, the grass can maintain its maximum photosynthetic gas exchange rates under water limitations [35]. These physiological advantages are further enhanced under elevated CO_2_ concentrations, which are locally present in urban and agricultural environments, effectively stimulating biomass and carbon accumulation by this species [22,36,37,38]. The carbon isotopic composition of C_4_ plants generally responds to water availability, with discrimination against ^13^C increasing with aridity [39,40,41]. Given that the underlying environmental characteristics of the collection sites were indistinguishable among land-use classes, the differences that we observed in the isotopic signatures of buffelgrass were assumed to be, at least in part, caused by local differences in gaseous carbon concentrations and sources. This has been especially documented in urban environments, where the intensive use of fossil fuel for vehicles and industry increases the local concentrations of anthropogenic CO and CO_2_, whose distinctive isotopic signatures can be tracked by the vegetation growing in these environments [11,17]. 

The lowest carbon content, which we found for plants collected from a small coastal locality that is mainly dedicated to tourism (Site no. 21), was accompanied by the most positive δ^13^C values of the study area, similar to the isotopic signatures for other C_4_ grasses in natural environments [42]. In turn, the highest carbon content was found at various sites within the Yaqui Valley, dominated by agricultural activities that presumably lead to a locally high CO_2_ concentration, not only from the use of agricultural machinery and the transportation of supplies, products, and workers, but also from biogenic emissions released by agricultural soil nutrient management, such as the use of synthetic fertilizers and, in many cases, the burning of crop residues [43,44,45]. The carbon compounds that are released from biomass burning also have isotopic signatures that are considerably more negative than those from “clean” environments, leaving a distinctive signature on the exposed grasses [18,21,46]. Additionally, the highest nitrogen emissions from the biomass burning typical of anthropic environments have a significant effect on the carbon content, as the increased nitrogen availability can increase photosynthetic rates [13,47]. 

The nitrogen concentration of plants growing in anthropic environments is strongly related to the amount of foliar uptake of reactive nitrogen species and to the rate of nitrogen deposition on the soil surface [13,48]. Plants growing close to a significant source of emissions such as traffic, industries, cultivated fields, or animal husbandry, have been found to increase considerably their nitrogen concentration, changing, from 0.8% in rural to 3.6% (dry weight) in highly urbanized areas [8,10,13]. Buffelgrass followed this pattern, with the lowest nitrogen concentration measured for individuals from a remote site with scant human activity, where nitrogenous emissions are presumably lower than for other sites included in this study, and the highest concentration found for plants collected near agricultural fields or major roads. In this respect, anthropogenic compounds, such as nitrogen oxides (NOx) and ammonia (NH_3_) are easily assimilated by plants via gas exchange (dry deposition). These compounds can also be dissolved into water to be deposited onto the soil by rain (wet deposition), where they can be taken up by the roots, further increasing the plant nitrogen concentration [13,14,49].

The nitrogen isotopic composition of the vegetation depends on different factors, with the regional distribution of human activity being one of the most important [14,50,51]. For example, plants growing in rural and natural sites, where either the nitrogen deposition rates are low (< 5 Kg N ha^−1^ year^−1^) or the nitrogenous emissions are low, tend to have δ^15^N values that are negative but close to zero [52]. In these areas, the main sources for nitrogen are lightning and biological fixation by legumes, which do not discriminate between isotopes. On the other hand, soil type and its particular isotope signature also play a large role in plant δ^15^N values, this is particularly important for soils in natural regions. However, when the nitrogen emissions and deposition increase, for example, as a consequence of traffic or agriculture, the importance of soil type and their original δ^15^N values for plants isotope compositions becomes reduced [14,53]. We observed this effect for the different land uses comprising our study area. In particular, as the human activity increased δ^15^N values became more positive. In addition, in natural regions the microbial and vegetation N cycling are closely coupled, i.e., no important modifications because land use or atmospheric deposition are presented and N cycling is “in equilibrium”, for these regions the leaching and gaseous N losses are negligible, thus the isotopic fractionation is low, and the δ^15^N values are generally negative close to zero [54,55,56,57]. 

The δ^15^N values that we measured for buffelgrass were consistently positive, regardless of the site of collection. For plants growing in urban environments, this has been attributed to the interaction of two main factors. First, urban vegetation is exposed to gaseous NOx which are taken up by direct stomatal absorption, these pollutants emitted by industry and cars are usually enriched in ^15^N, with δ^15^N values that can reach up to 10‰, which explains in part the isotopic signature that we found for buffelgrass in those environments [10,14,47,49,51]. The NOx can be dissolved in water as nitrate (NO_3_^−^) and deposited either by rain, fog, or snow onto the soil and reach the urban vegetation, their δ^15^N values are generally positive [54,58]. An additional source of nitrogen is reduced gaseous NH_3_ or dissolved NH_4_^+^, compounds that tend to be ^15^N depleted [54]. However, reduced nitrogen compounds do not abound in cities, thus representing a small proportion of the nitrogen sources for urban vegetation [10,54]. Second, urban vegetation, especially ruderal plants such as buffelgrass, have positive δ^15^N values because they root on modified soils, in which the nitrogen reactions are altered leading to a ^15^N enrichment as discussed below [44,57,58,59,60,61,62]. Therefore, the combination of the constant stomatal uptake of oxidized nitrogen by leaves and roots, coupled with rooting on modified soils contributed to the observed positive δ^15^N values for buffelgrass in the urban environments [14]. As it occurs for plants from urban environments, those growing near roads and highways and even from isolated sites are exposed to the same process, but to a lesser extent. This explains why the δ^15^N values for buffelgrass changed in the human activity gradient from very positive in urban to less positive closer to zero in isolated regions where emissions are low.

For agricultural areas, the nitrogenous fertilizer applied to crops can be taken up by plants, released as a gas, retained in the soil, or leached to the groundwater. For the case of the Yaqui Valley in our study region, one third of the applied nitrogen ends up in harvested crops, the other two thirds are released to the environment [63,64]. Not only for agricultural lands, but also for areas with other land uses, increasing rates of deposition result in soil nitrogen losses, which produces the ^15^N enrichment of the available nitrogen in soil for plants [14,53,65,66,67,68]. For example, N_2_O emissions are produced in agricultural fields after fertilization and in ornamental lawns when the incomplete transformation of NH_4_^+^ to nitrate, or during the incomplete transformation of NO_3_^−^ to N_2_ occurs, nitrification or denitrification, respectively. N_2_O emissions can reach δ^15^N values as negative as −48.2‰ [68,69]. NH_3_ volatilization is another form of nitrogen loss from agricultural and other soils exposed to increasing rates of nitrogen inputs; it has δ^15^N values that reach −56‰ [70,71].

Both N_2_O and NH_3_ can be directly absorbed by plants growing near agricultural fields, which would result in very negative δ^15^N values [50,58]. Another potential N source with negative δ^15^N values is NO_3_^–^, which is massively used in the study region for crop fertilization, such that it is lixiviated in high quantities, e.g., 8200 Mg N/yr just in the Yaqui Valley, most of which reaches the groundwater and is exported to the Gulf of California [63,72], similar to that which occurs in other agricultural lands [64]. However, such isotopic signals were not detected here. Considering that fertilizers are applied directly to the crops and we collected in the vicinity, but not directly in the agricultural plots, the positive δ^15^N values that we measured for buffelgrass in agricultural sites may have multiple origins. For instance, when enough nitrogen accumulates in the soil, as a consequence of high rates of nitrogen deposition, gaseous reactive species of nitrogen (NOx or NH_3_) have been shown not to have a significant effect on the δ^15^N value of grasses regardless of their local atmospheric concentration [13,50]. Additionally, the ^15^N enrichment that results from N loss is common in disturbed soils, such as those where we collected buffelgrass [13,53,65,66,67,73]. In addition, the high rates of deposition of NOx-derived compounds that are prevalent in many areas with anthropogenic activities may help explain the positive δ^15^N values found in the study area [74,75]. 

The isotopic differences that we found among the various land-use classes may be influenced, at least in part, by existing differences in the isotopic fractionation rates of soil processes involved in nitrogen loss [64,65,66], which in turn respond to the amount and type of available nitrogen. The signal of the nitrogen remaining in the soil is picked up by the plants. However, higher rates of nitrogen deposition do not necessarily result in higher carbon or nitrogen content in plant tissues because negative feedback mechanisms can reduce or even halt further assimilation of nitrogen beyond species-specific physiological thresholds, which have been documented for various epiphytic orchids, terrestrial herbs, and trees [8,47,66,67,76]. Whether the amount of nitrogen available in the study region surpassed such a threshold for buffelgrass should be considered in future work by determining both the nitrogen-tolerance of this species and conducting measurements of the prevailing forms of nitrogen available in the various atmospheric and edaphic compartments. Our results, however, provide some indication of the regions in Sonora that are more susceptible of reaching such a nitrogen saturation threshold, including the Yaqui Valley, Coast of Hermosillo, and the two large cities of the region.

As the geographical expansion of buffelgrass continues in Mexico, owing to its tolerance to high temperatures and low water availability, its use as a biomonitor of nitrogen deposition can be a silver lining for this biological invasion as the grass’ elemental and isotopic composition appears to respond directly to the intensity of anthropogenic disturbance under different land uses.

## 4. Materials and Methods

### 4.1. Study Region

The study was conducted in the state of Sonora, located in northwestern Mexico, along the Gulf of California (Figure 1). Collection sites were selected by the spontaneous occurrence of buffelgrass in this heterogeneous landscape. Each site was assigned to one of five classes of land use along a gradient of intensity of human activity [77]: (1) urban settlements from 1 to 49,999 habitants (Small urban), (2) urban settlements from 50,000 to 399,000 (Medium urban), (3) settlements more than 400,000 (Large urban), (4) agriculture and, (5) rural landscape. The study area included two important agricultural regions, the Yaqui Valley (6 sites), accredited to be the birthplace of the Green Revolution, comprises ca. 230,000 ha of irrigated cropland, mainly for the production of wheat and maize [74]. The coast of Hermosillo (10 sites), with 1.5 million ha, also under irrigation, is mainly dedicated to the production of various horticultural crops, such as table grapes and watermelon. The study region also contained two major cities, Hermosillo (population 812,229) and Ciudad Obregón (433,050), as well as various mid-sized settlements [28,78]. 

Climate and soil characteristics were determined for each collection site. In particular, the minimum temperature of the coldest month, the maximum temperature of the warmest month, the mean monthly temperature, and the annual precipitation were obtained for the 1961–2001 normals, through climatic spline models for each site [79]. In turn, the soil’s geological origin, primary soil type, clay, silt, and sand content, electrical conductivity, and pH were obtained for the nearest reference point from the National soil profile database [80]. 

### 4.2. Sampling of Plant Material

Tissue samples of the invasive exotic grass *Cenchrus ciliaris* L. 1771, which is amply distributed in the study region [22,23], were collected from 34 sites from Sonora, Mexico, in March 2017. In particular, fully developed green leaves and mature seeds (fascicles were left intact) were collected at each site from each one of 5 individuals growing at least 5 m apart. The samples were placed in paper envelopes and kept in a shaded cardboard container during its transport to the laboratory, which usually occurred within 2 days.

### 4.3. Elemental and Isotopic Analysis

The plant materials were dried at 60 °C in a gravity convection oven until reaching constant weight, usually within 48 h. The dried tissues were ground to a fine powder in a ball mill, wrapped into tin capsules (Costech Analytical, Inc. Valencia, California, USA), and weighed with a microbalance (resolution of 0.01 mg, Sartorius, Götingen, Germany). For each sample, the carbon and nitrogen content, as well as the corresponding isotopic composition, were determined at the Stable Isotope Facility University of Wyoming (Laramie, WY, USA). The analyses were conducted with a Costech 4010 elemental analyzer (Costech Analytical Inc., Valencia, CA, USA) attached to a continuous flow isotope ratio mass spectrometer (Finnigan Delta Plus XP, Thermo Electron Corp, Waltham, USA). Carbon and nitrogen isotope ratios, reported in parts per thousand, were calculated relative to V-PDB or atmospheric air by means of glutamic acid and alfalfa secondary standards (accessions: 36-UWSIF-Glutamic 1, 39-UWSIF-Glutamic 2, and UWSIF05). The analytical precision for the δ^13^C was 0.3 ± 0.02‰ and 0.4 ± 0.03‰ (SD) for δ^15^N. The natural abundances of ^13^C and ^15^N were calculated as:δ^13^C (‰ _versus V-PDB_) = (R_sample_/R_standard_ − 1) × 1000
δ^15^N (‰ _versus air_) = (R_sample_/R_standard_ − 1) × 1000
where R is the ratio of ^13^C/^12^C for carbon and ^15^N/^14^N for nitrogen isotope abundance for a given sample [81,82].

### 4.4. Statistical Analyses

To determine whether an underlying grouping of soil and climate characteristics existed for the five land-use categories, we conducted a Principal Components Analysis, excluding the categorical attributes of soil geological origin and primary type, followed by a Kruskal–Wallis test by ranks for the values of the collection sites along each principal component.

Differences among land use and organs for the plant carbon and nitrogen contents, the C:N ratio, as well as for the δ^13^C and δ^15^N values, were determined by means of a non-parametric two-way PERMANOVA (factors were organ and land use class). Pairwise comparisons were made by means of the function pairwise.adonis 2 from the ‘Vegan’ package in R (*p* < 0.05). Analyses were conducted in R (version 3.5.3, R Core Team, R foundation for Statistical Computing, Vienna, Austria), except for the Principal Components Analysis that was run on ClustVis [83].

### 4.5. Geostatistical Analysis

The ordinary Kriging method [84] within ArcGIS 10.5 (Esri, Redlands, California, USA), was used to determine the spatial distribution of plant carbon and nitrogen content, the C:N ratio, and the δ^13^C and δ^15^N values based on the interpolation of the point data measured for buffelgrass.

## Figures and Tables

**Figure 1 plants-10-00942-f001:**
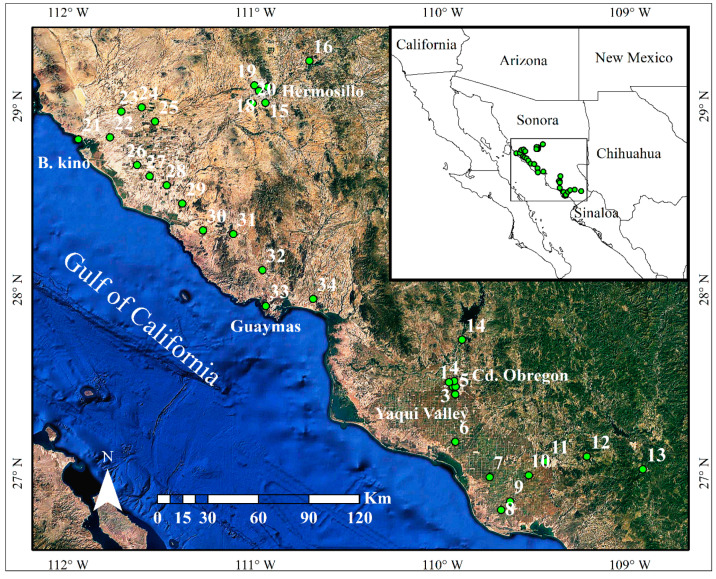
Study area in Sonora, Mexico. Green circles indicate the sites where buffelgrass was collected. Numbers indicate the collection site from small (sites 13 and 21), medium (9, 33 and 11) and large urban areas (1–4, 15, 17–19), agricultural (5–8, 10, 14, 16, and 24–30) and rural sites (20, 22, 23, 31, 32, and 34). Image data: Google Earth; date: 20 April 2020.

**Figure 2 plants-10-00942-f002:**
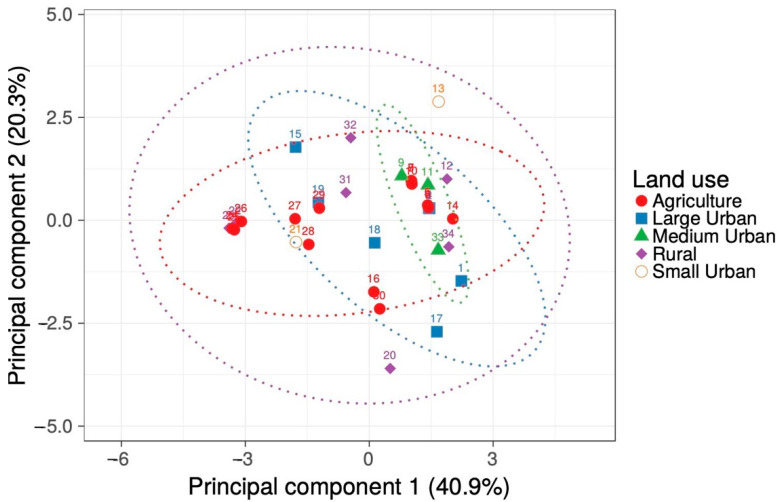
Principal Components Analysis for climate and soil characteristics from 34 collection sites with different land use. Data are plotted along the two main components from Table 2 for sites with agricultural (circles), large (squares), medium (triangles), and small (open circles) urban, or rural (diamonds) use. The numbers are the site identity from Figure 1. Ellipses indicate the 95% confidence interval for each land-use class. Percentages along the axes indicate the variance explained by each component.

**Figure 3 plants-10-00942-f003:**
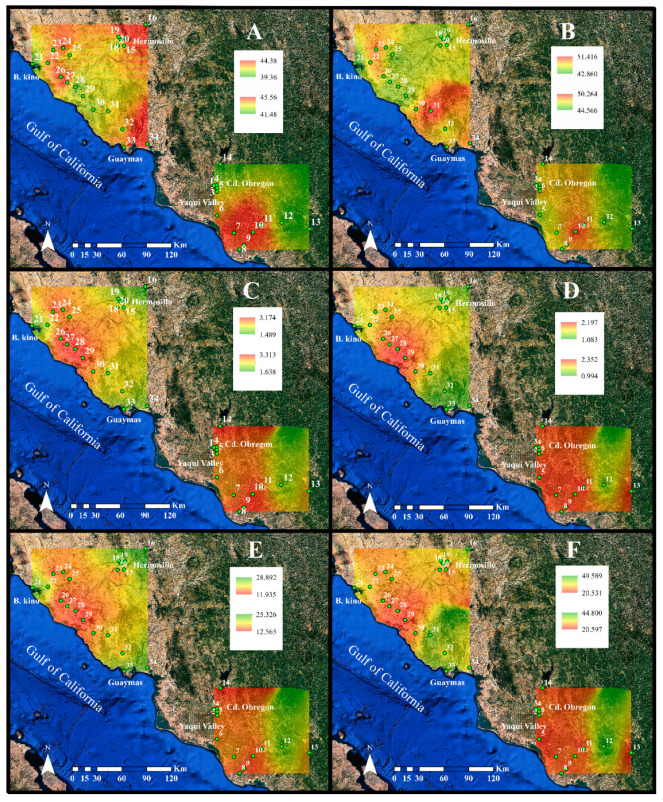
Spatial distribution for the elemental composition for buffelgrass in Sonora, Mexico. Carbon content (C%, dry weight) for (**A**) leaves and (**B**) seeds, nitrogen content (N%, dry weight) for (**C**) leaves and (**D**) seeds, and the C:N ratio for (**E**) leaves and (**F**) fascicles. Image data: Google Earth; date: 20 April 2020.

**Figure 4 plants-10-00942-f004:**
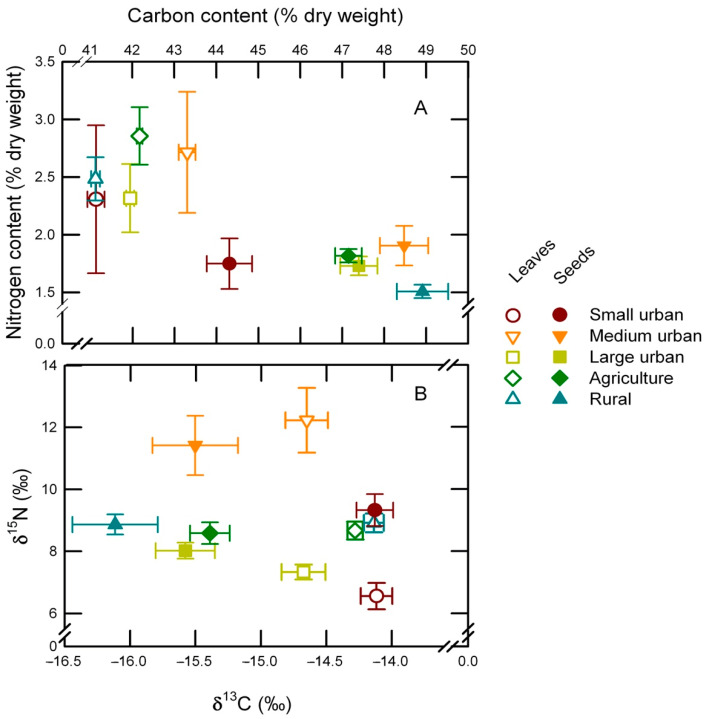
Elemental and isotopic carbon and nitrogen composition for leaves and seeds of *C. ciliaris* collected from sites with different land use. (**A**) Carbon and nitrogen content (dry mass basis) for leaves (open symbols) and seeds (closed symbols) collected from small (circles), medium (inverted triangles), and large cities (squares), agricultural lands (diamonds), and rural sites (triangles). (**B**) Carbon and nitrogen isotopic signatures for leaves and seeds of *C. ciliaris* collected from sites with different land use. Data are shown as mean ± S.E. See underlying data in Appendix A.

**Figure 5 plants-10-00942-f005:**
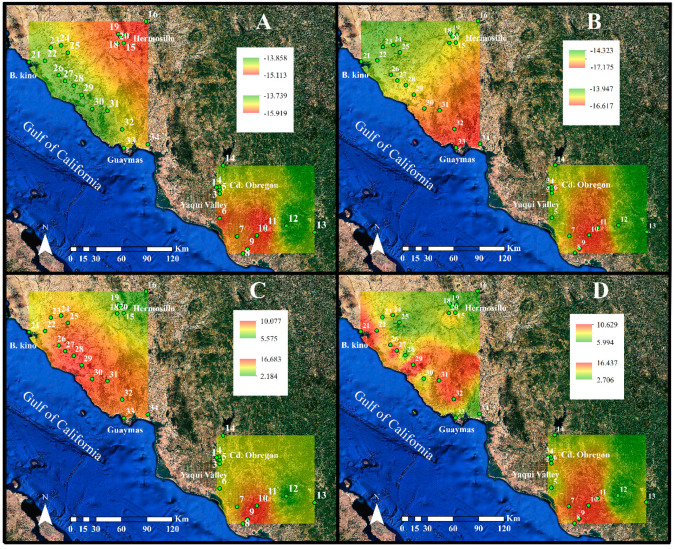
Spatial distribution for buffelgrass isotopic composition in Sonora, Mexico. Carbon isotopic composition (δ^13^C, ‰) for (**A**) leaves and (**B**) seeds and nitrogen isotopic composition (δ^15^N, ‰) for (**C**) leaves and (**D**) seeds. Image data: Google Earth; date: 20 April 2020.

**Table 1 plants-10-00942-t001:** Climate and soil attributes from the study region in Sonora, Mexico. Data are means ± 1 S.E. (n = 34 sites).

Parameter	Value
Mean temperature (°C)	23.5 ± 0.1
Minimum temperature (°C)	7.5 ± 0.2
Maximum temperature (°C)	38.0 ± 0.1
Annual precipitation (mm)	287 ± 20
Primary soil type	Cambisol (26% of sites), luvisol (26%), vertisol (24%), leptosol (12%), regosol (6%), phaeozem (6%)
Geological origin	Alluvial (85%), sedimentary (12%), igneous (3%)
Clay content (%)	32.2 ± 3.0
Silt content (%)	22.5 ± 2.0
Sand content (%)	45.3 ± 3.6
Electrical conductivity (mmho/cm)	0.71 ± 0.14
pH	8.45 ± 0.14

**Table 2 plants-10-00942-t002:** Principal Components Analysis for climate and soil attributes from 34 sites in Sonora, Mexico. Data are the component loadings for each attribute considered and the percent of the variance explained by each component individually and cumulative. For each component, the *p*-values are from a Kruskal–Wallis test for the axis values for each land use class.

	Principal Component								
	1	2	3	4	5	6	7	8	9
Sand	0.32	0.39	−0.28	0.14	−0.35	−0.28	0.29	−0.04	0.59
Silt	0.03	0.38	0.37	−0.55	0.47	0.19	0.05	0.07	0.39
Clay	−0.29	−0.54	0.03	0.19	0.04	0.13	−0.27	−0.01	0.70
Minimum temperature	−0.51	0.21	0.06	−0.13	−0.13	−0.27	−0.09	−0.76	−0.00
Maximum temperature	0.13	−0.39	−0.63	−0.44	0.32	−0.06	0.27	−0.24	−0.00
Mean Temperature	−0.49	0.03	−0.13	−0.24	−0.02	−0.58	−0.01	0.58	0.00
Precipitation	−0.35	0.30	−0.47	−0.15	−0.29	0.65	−0.12	0.14	0.00
Electrical conductivity	0.40	0.03	−0.13	−0.31	−0.16	−0.18	−0.82	−0.02	−0.00
pH	−0.07	0.35	−0.35	0.50	0.65	−0.09	−0.27	−0.02	−0.00
Individual variance	0.41	0.20	0.13	0.12	0.07	0.04	0.03	0.00	0.00
Cumulative variance	0.41	0.61	0.74	0.86	0.93	0.97	1.00	1.00	1.00
*p*-value	0.328	0.812	0.036	0.232	0.812	0.079	0.814	0.085	1.00

**Table 3 plants-10-00942-t003:** Two-way PERMANOVA for carbon and nitrogen parameters of buffelgrass growing in Sonora, Mexico. *p*-values in **bold** indicate significant differences.

		C Content (% Dry Weight)		N Content (% Dry Weight)		C:N Ratio		δ^13^C (‰)		δ^15^N (‰)	
	d.f.	F	*p*	F	*p*	F	*p*	F	*p*	F	*p*
Organ	1	463.2	**0.001**	172.3	**0.001**	276.98	**0.001**	94.641	**0.001**	0.741	0.416
Land use	4	6.261	**0.001**	5.109	**0.002**	4.529	**0.001**	4.892	**0.002**	12.89	**0.001**
Organ × Land use	4	4.805	**0.002**	2.285	0.057	2.918	**0.015**	4.037	**0.003**	2.954	**0.012**

## Data Availability

Data are available from the corresponding author on reasonable request.

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
