# Peer review of "Influence of Land Use on the C and N Status of a C4 Invasive Grass in a Semi-Arid Region: Implications for Biomonitoring"

_plants, 2021, doi:10.3390/plants10050942_

Round 1

Reviewer 1 Report

General comment

The manuscript by Diaz-Alvarez and de la Barrera reports on an approach to using an invasive grass species (Cenchrus ciliaris, C4 photosynthesis) as biomonitor for carbon and nitrogen atmospheric pollution in a semi-arid environment in NW Mexico. The authors collected in five replicates each green leaves and mature seeds from 34 locations in Sonora, Mexico. The locations included remote sites, pastures and other agricultural sites as well as highway-affected sites and sites within cities. The collected plant material was analysed for carbon and nitrogen concentrations, C:N ratios and carbon and nitrogen stable isotope natural abundance. Differences among sites and plant compartments in the analysed parameters were tested for significance by ANOVA. Standard geostatistical analysis was used to interpolate between sampling locations and to produce spatial distribution maps.

The authors found considerable differences in their parameters between sites and claim that these differences are driven by pollutant depositions due to anthropogenic activities and that the investigated grass species is suited to serve as a biomonitor for atmospheric pollution. This claim may hold true or not. In any case, anthrophogenic pollution is much more complex than the authors make the reader belief. While monitoring of atmospheric carbon pollution is relatively easy due to a preferential uptake of only one carbon source, atmospheric CO2, the situation for monitoring atmospheric nitrogen pollution is much more complicated. Atmospheric nitrogen deposition can be taken up by aboveground plant compartments as gaseous deposition and as wet deposition. Gaseous and wet deposition can be composed of oxidized and reduced nitrogen compounds with essentially different anthropogenic origins and nitrogen isotope signatures. Furthermore, wet nitrogen deposition also reaches the soil. From the soil, nitrogen deposition can be taken up by plant roots, however, nitrogen deposition can also affect the entire nitrogen cycle in soils, e.g. by nitrate leaching, denitrification, increased nitrification etc., and thus, can affect the soil-borne nitrogen isotope composition. All these factors have to be taken into account when claiming a soil-rooted plant as biomonitor for atmospheric nitrogen deposition. I urgently recommend reading and considering a whole bunch of literature on this point: Gebauer & Schulze (1991) Oecologia 87: 198-207; Gebauer & Dietrich (1993) Isot. Envir. Heath Stud. 29: 35-44; Gebauer et al. (1994) Plant Soil 164: 267-281; Jung et al. (1997) Envir. Pollut. 97: 175-181; Harrison et al. (2000) Ecol. Stud. Vol. 142: 171-188; Bauer et al. (2000) Ecol. Stud. Vol. 142: 189-214; Parra Suarez et al. (2019) STOTEN 638: 66-79.

Specific comments

Line 28 and throughout the manuscript: “Anthropic”? Do you mean anthropogenic?

Line 39: Please note that sulfur is no atmospheric pollutant. I suspect you mean sulfur dioxide and/or sulfate.

Lines 144-145: This is an incomplete sentence. Please correct.

Lines 196-203: What is about nitrogen fertilizers and their effects on non-agricultural sites?

Line 211: Please note that NOx and NH3 usually have essentially contrasting isotope signatures. While NOx trends to be enriched in 15N, NH3 can be heavily depleted in 15N.

Lines 257-270: (1) I am missing a description on how you used an EA-IRMS coupling to come up with carbon and nitrogen concentrations. (2) Please note that the Carlo Erba EA 1110 elemental analyser was not produced by Costech Analytical, but rather by Carlo Erba, Milano, Italy. (3) Linking stable isotope abundance data to international standards requires always a measurement of secondary standards for calibration. Which secondary standards were used in your investigation?

Lines 278-285: The use of the parametric test procedure ANOVA requires a normal distribution and homogeneity of variances in your dataset. According to my own experience, stable isotope natural abundance data are in most cases non-normally distributed. Thus, please check whether the requirements for a use of ANOVA are really fulfilled in your case or alternatively use non-parametric test procedures.

Author Response

Reviewer 1

Open Review

(x) I would not like to sign my review report

( ) I would like to sign my review report

English language and style

( ) Extensive editing of English language and style required

( ) Moderate English changes required

(x) English language and style are fine/minor spell check required

( ) I don't feel qualified to judge about the English language and style

Yes

Can be improved

Must be improved

Not applicable

Does the introduction provide sufficient background and include all relevant references?

( )

(x)

( )

( )

Is the research design appropriate?

( )

(x)

( )

( )

Are the methods adequately described?

( )

( )

(x)

( )

Are the results clearly presented?

( )

(x)

( )

( )

Are the conclusions supported by the results?

( )

( )

(x)

( )

Comments and Suggestions for Authors

General comment

The manuscript by Diaz-Alvarez and de la Barrera reports on an approach to using an invasive grass species (Cenchrus ciliaris, C4 photosynthesis) as biomonitor for carbon and nitrogen atmospheric pollution in a semi-arid environment in NW Mexico. The authors collected in five replicates each green leaves and mature seeds from 34 locations in Sonora, Mexico. The locations included remote sites, pastures and other agricultural sites as well as highway-affected sites and sites within cities. The collected plant material was analysed for carbon and nitrogen concentrations, C:N ratios and carbon and nitrogen stable isotope natural abundance. Differences among sites and plant compartments in the analysed parameters were tested for significance by ANOVA. Standard geostatistical analysis was used to interpolate between sampling locations and to produce spatial distribution maps.

The authors found considerable differences in their parameters between sites and claim that these differences are driven by pollutant depositions due to anthropogenic activities and that the investigated grass species is suited to serve as a biomonitor for atmospheric pollution. This claim may hold true or not. In any case, anthropogenic pollution is much more complex than the authors make the reader believe. While monitoring of atmospheric carbon pollution is relatively easy due to a preferential uptake of only one carbon source, atmospheric CO2, the situation for monitoring atmospheric nitrogen pollution is much more complicated. Atmospheric nitrogen deposition can be taken up by aboveground plant compartments as gaseous deposition and as wet deposition. Gaseous and wet deposition can be composed of oxidized and reduced nitrogen compounds with essentially different anthropogenic origins and nitrogen isotope signatures. Furthermore, wet nitrogen deposition also reaches the soil. From the soil, nitrogen deposition can be taken up by plant roots, however, nitrogen deposition can also affect the entire nitrogen cycle in soils, e.g. by nitrate leaching, denitrification, increased nitrification etc., and thus, can affect the soil-borne nitrogen isotope composition. All these factors have to be taken into account when claiming a soil-rooted plant as biomonitor for atmospheric nitrogen deposition. I urgently recommend reading and considering a whole bunch of literature on this point: Gebauer & Schulze (1991) Oecologia 87: 198-207; Gebauer & Dietrich (1993) Isot. Envir. Heath Stud. 29: 35-44; Gebauer et al. (1994) Plant Soil 164: 267-281; Jung et al. (1997) Envir. Pollut. 97: 175-181; Harrison et al. (2000) Ecol. Stud. Vol. 142: 171-188; Bauer et al. (2000) Ecol. Stud. Vol. 142: 189-214; Parra Suarez et al. (2019) STOTEN 638: 66-79.

We appreciate this observation and, in consequence, have de-emphasized the biomonitoring component and rather refocused on the distribution of C and N. We also included a new example (Redling et al. 2013) of grass biomonitors that adds on to the two articles that we had included in the original version (Wang and Pataki 2010, 2011).

As this reviewer points out, air pollution is a very complex phenomenon that includes several stages of transformation and deposition, including plant-atmosphere and soil-atmosphere interactions. We appreciate the recommended literature and have included most of these references in the revised manuscript. It was most helpful in improving our explanation about nitrogen deposition and its effects on the isotope composition of plants, including the processes in different compartments. Specifically, the revised discussion now includes an explanation of various factors influencing the isotopic composition of plants exposed to different rates of atmospheric nitrogen deposition. The main argument for the use of buffelgrass as a biomonitor of atmospheric nitrogen deposition is based on the fact that as the rate of deposition increases, the biogeochemical properties of soil change, including changes in the rates of nitrification and denitrification, which in turn lead to higher soil nitrogen loses, and a highly enriched soil (with 15N). In this case, a higher the rate of deposition leads to δ15N values.

Specific comments

Line 28 and throughout the manuscript: “Anthropic”? Do you mean anthropogenic?

We generally meant anthropic, i.e., related to humans. "Anthropic" appeared six times in the original manuscript, we changed it to "anthropogenic" when referring to carbon and nitrogen emissions in the conclusions. In the other instances, anthropic qualifies "activities" (one time) and "environments", which are not necessarily created by humans. In turn, "anthropogenic" appeared twice to qualify land use change and the emission of pollutants.

Line 39: Please note that sulfur is no atmospheric pollutant. I suspect you mean sulfur dioxide and/or sulfate.

Indeed, we meant SO2 and the ensuing sulfate aerosols and (sulfuric) acid rain. We have specified SO2 in the revised manuscript, which is the gas that is directly measured by monitoring stations.

Lines 144-145: This is an incomplete sentence. Please correct.

Thank you for your noticing this edition oversight. We have reworded this sentence for clarity.

Lines 196-203: What is about nitrogen fertilizers and their effects on non-agricultural sites?

We have indicated in the revised discussion that "Two thirds of the nitrogenous fertilizers applied to croplands in the Yaqui Valley are lost, mainly by the volatilization of N2O or NH3. NO2 contributes to climate change and the NH3 can become part of the atmospheric nitrogen deposition, which in turn can reach the neighboring vegetation including buffelgrass."

Line 211: Please note that NOx and NH3 usually have essentially contrasting isotope signatures. While NOx trends to be enriched in 15N, NH3 can be heavily depleted in 15N.

This observation by the reviewer is true. Considering that nitrogen deposition is composed of different compounds, with different origins and isotope compositions, we have included a description of various sources and their respective δ15N values in the revised discussion.

Lines 257-270: (1) I am missing a description on how you used an EA-IRMS coupling to come up with carbon and nitrogen concentrations. (2) Please note that the Carlo Erba EA 1110 elemental analyser was not produced by Costech Analytical, but rather by Carlo Erba, Milano, Italy. (3) Linking stable isotope abundance data to international standards requires always a measurement of secondary standards for calibration. Which secondary standards were used in your investigation?

1) The University of Wyoming Stable Isotope Facility provides this explanation for the carbon and nitrogen contents in plant tissue, which is conducted by the elemental analyzer: "Carbon and nitrogen containing compounds occur in both organic (such as plants and animals) and inorganic (such as soils) systems.  δ13 and δ15N ratios of the material can be determined by isotopic analysis of the CO2 and N2 generated by the combustion of the material. At SIF, the homogenized sample is accurately weighed into a tin capsule and loaded into the autosampler attached to an elemental analyzer.  The EA is coupled to an IRMS through an open split interface.  When a sample is dropped into the EA, the EA automatically processes the sample, combusting the sample to generate product gases that are then separated using the gas chromatograph.  The separated gases (CO2 and N2) are then pass through the open split interface to the IRMS for isotopic analysis (EA-IRMS)." Please see diagram at http://www.uwyo.edu/sif/instrumentation/elemental-analyzer.html

2) Indeed, the elemental analyzer was actually a Costech 4010. This has been corrected in the revised manuscript.

3) We have specified in the revised manuscript that two glutamic acid and an alfalfa secondary standards were utilized in the stable isotopic analyses. We have also included the SIF accession numbers.

Lines 278-285: The use of the parametric test procedure ANOVA requires a normal distribution and homogeneity of variances in your dataset. According to my own experience, stable isotope natural abundance data are in most cases non-normally distributed. Thus, please check whether the requirements for a use of ANOVA are really fulfilled in your case or alternatively use non-parametric test procedures.

As the reviewer suggested, we revised our statistical analysis. In the revised manuscript we utilized the more appropriate non-parametric two way PERMANOVA analysis to identify differences among the various land uses.

Reviewer 2 Report

This manuscrpt by Diaz-Alvarez (apologies for the lack of accents) and de la Barrera tests the suitability of an invasive C4 grass (bufful grass) as a biomonitor for C and N atmospheric pollution.  The work is an interesting concept, and the authors are talented writers.

However, I do have a few concerns, some of them quite major.

1) I think there needs to be considerably more data transparency.  For example, the plant characteristics (%N, %C, isotopic analysis) should be presented in a table (along with errors, replicate numbers) with a description of each site (with site types grouped, and possibly means presented for each site group).  The results talked really about the range of results (max, min), but I didn't get much of a feel for the data, and I couldn't inspect it myself, since it wasn't presented.

2) I think the sites need to be characterised better (i.e. temperature, rainfall, soil types, dominant usage).  When you talk about rural vs urban sites, it is implied that C / N deposition are the main factors, while it may be that urban areas tend to develop in areas nearer water bodies, or with soil characteristics more appropriate for agricultural development.  It seems plausible that your results are an artifact of other factors which haven't been explored.

3) I think some monitoring of atmospheric conditions is probably also appropriate - for example, CO2 levels, NOx levels, etc.  Can you show a correlation between the atmospheric conditions and the plants?  In order to use your plants for the stated purpose, that would seem to be vital.

4) I have some concerns about the GIS map data presented.  Although the technique seems in principle useful, there are some pretty big gaps between some clustered sites (e.g. sites along the coast from B. kino to Guaymas vs. the in-land sites arounf Hermosillo), so the model would be quite unreliable in the area between them.  Perhaps tabulating your data into groups and looking for commonality within sites and difference between sites would be more appropriate.

5) In Table 1, I think you need to look again at some of the reported p-values.  For example, I am not sure that where F=3.486, p equals <0.001.  MS Excel returns a p of 0.071 for F=3.486, and df. of 1, 33.  It may be your analysis is a bit different from my "back of the envelope" calculations, but I would suggest to check them all.

6) End of Introduction.  Since this journal style runs Intro --> Results, I think you need a paragraph at the end of the Intro which sets out what you will do, and how your information will be structured.  For example
"In this study, in order to test xyz, we measured core physiological (%C, %N) and isotopic (13C, 15N) parameters of buffel grass tussocks sampled from 30 sites in the north west of Mexico" or something like this.

Minor

Sometimes the writing is a little slack.  For example, the first sentence of the Introduction tends to run-on, and should be split into two sentences.  "Extensions of territory" (ln40) could be simplied to "area".  (indeed, I feel this whole sentence is backwards in structure)

Ln57 - "In this respect" -- I'm not sure that you mean. Perhaps you mean "To address this limitation..." or similar?

Ln 71 - "and their intensive and polluting activity" - this feels out of place.  It also feels like something which needs some evidence regarding local conditions.

Ln114: "28.8 f"

Ln120;  "were as low as -15.9‰"  --> "ranged from -15.9‰ ... .... to -13.7‰"

Ln127: "cultivation fields" --> cultivated fields

The discussion was rather well written, but I felt it was a little more hypothetical, rather than dealing with the phenomena in your data.  I take issue with a couple of points.

1) The fact that Buffel grass is C4 should have been brought up in the intro, not the discussion

2) Ln150/1 -- discussing isotopic discrimination with respect to aridity "which was not oberved here" -- you never show us this in the data -- it's just an assertion. 

3) Ln215 - you talk about soil saturation.  Did you measure it?  Where was that reported in the data?  I didn't see it.

4) Ln220 -- "the likely result" -- how did you determine this was the most likely result?  I could not do that based on what you showed us! 

5) Ln226 -- "threshold of saturation" -- my review notes just say "prove it".  I don't see any evidence of this in your manuscript.  

Overall, I think you need to present more of your data, so that readers can understand and draw their own conclusions.  At the moment, it feels like you're just telling us a story, and we are simply required to believe you.

I apologise if my comments seem strong.  However, I do think you have interesting data, but I am currently unable to put your numbers into any context, and it's very frustrating.

Author Response

Reviewer 2

Open Review

(x) I would not like to sign my review report

( ) I would like to sign my review report

English language and style

( ) Extensive editing of English language and style required

(x) Moderate English changes required

( ) English language and style are fine/minor spell check required

( ) I don't feel qualified to judge about the English language and style

Yes

Can be improved

Must be improved

Not applicable

Does the introduction provide sufficient background and include all relevant references?

( )

(x)

( )

( )

Is the research design appropriate?

( )

( )

(x)

( )

Are the methods adequately described?

( )

( )

(x)

( )

Are the results clearly presented?

( )

( )

(x)

( )

Are the conclusions supported by the results?

( )

( )

(x)

( )

Comments and Suggestions for Authors

This manuscript by Diaz-Alvarez (apologies for the lack of accents) and de la Barrera tests the suitability of an invasive C4 grass (buffel grass) as a biomonitor for C and N atmospheric pollution.  The work is an interesting concept, and the authors are talented writers.

However, I do have a few concerns, some of them quite major.

We thank the reviewer for their kind assessment of the interest of the work and the delivery. We do appreciate their concerns, which we have attempted to address, as explained below in detail.

1) I think there needs to be considerably more data transparency.  For example, the plant characteristics (%N, %C, isotopic analysis) should be presented in a table (along with errors, replicate numbers) with a description of each site (with site types grouped, and possibly means presented for each site group).  The results talked really about the range of results (max, min), but I didn't get much of a feel for the data, and I couldn't inspect it myself, since it wasn't presented.

We apologize for this lack of transparency, which was also noted by the other reviewer. It was not intentional. In the revised version we have included a new Figure 4 of plant responses grouped by land-use (and an amended Table 1 that now reports a 2-way PERMANOVA, with land use and plant organ as factors). This should give readers a better feel of the plant characteristics in the various land-use types. We also reclassified the land-use classes so that they are compatible with the Forman & Godron (1986) natural-urban use-intensity gradient.

2) I think the sites need to be characterised better (i.e. temperature, rainfall, soil types, dominant usage).  When you talk about rural vs urban sites, it is implied that C / N deposition are the main factors, while it may be that urban areas tend to develop in areas nearer water bodies, or with soil characteristics more appropriate for agricultural development.  It seems plausible that your results are an artifact of other factors which haven't been explored.

Indeed, all of these factors influence plant C and N status. In the revised manuscript we have considered both climatological and edaphic characteristics for each study site, as suggested by the reviewer. A principal components analysis revealed that the environmental conditions were indistinguishable among the five land-use classes. We can thus conclude that carbon and nitrogen emissions played an important role in the elemental and isotopic composition for buffel grass in the study area.

3) I think some monitoring of atmospheric conditions is probably also appropriate - for example, CO2 levels, NOx levels, etc.  Can you show a correlation between the atmospheric conditions and the plants?  In order to use your plants for the stated purpose, that would seem to be vital.

We completely agree with the reviewer that pollutant measurements are required to actually calibrate buffel grass as a biomonitor. Unfortunately, such data are not available for the study region. For example, the Mexican government's air quality information system indicates that monitoring exist in the cities of Ciudad Obregón (https://sinaica.inecc.gob.mx/estacion.php?estId=185) and Hermosillo (https://sinaica.inecc.gob.mx/estacion.php?estId=186), but they are not under operation due to a lack of personnel. It is precisely this lack of information on atmospheric pollution the driver of our research. (In fact, we estimate that upwards of 20 million people live in unmonitored Mexican cities, despite it being required by the country's environmental regulations).

We would also like to offer three examples of the use of a uncalibrated biomonitors: Bryologist (2021) 124:52–67, Chemosphere (2010) 78: 965-971, and J. Geophysical Res. (2011) 116: D24301. The latter work even utilized previously published works from various geographical origins to compile equations of plant status vs. environmental pollution to estimate pollution levels in China.

4) I have some concerns about the GIS map data presented.  Although the technique seems in principle useful, there are some pretty big gaps between some clustered sites (e.g. sites along the coast from B. kino to Guaymas vs. the in-land sites around Hermosillo), so the model would be quite unreliable in the area between them.  Perhaps tabulating your data into groups and looking for commonality within sites and difference between sites would be more appropriate.

We appreciate the reviewer's concern. We hope that with the revised Table 3 and the new Figure 4 of land-use vs. plant responses, the grouping requirement will be satisfied. We would insist, however, in keeping the maps, taking into account that the first purpose of biomonitoring is the spatial representation for pollution. The interpolation technique utilized in our study has been amply utilized for nitrogen and carbon emissions and the resulting model serves as an illustrative proxy for the state of pollution, which can be useful for policy makers.

Although there are no collection sites between Hermosillo and the coast, the model utilized for representing the distribution of pollutants in the landscape takes into account these particularities of the anthropic environments. In addition, gaps in our study are comparable to gaps from similar studies. Including Wang and Pataki (2009,2010) and our our own studies, for example, refs 10 and 12.

5) In Table 1, I think you need to look again at some of the reported p-values.  For example, I am not sure that where F=3.486, p equals <0.001.  MS Excel returns a p of 0.071 for F=3.486, and df. of 1, 33.  It may be your analysis is a bit different from my "back of the envelope" calculations, but I would suggest to check them all.

As indicated above, we actually redid the statistical analysis. We have now utilized the more appropriate non-parametric, two-way PERMANOVA (factors were land-use and plant organ) to identify possible differences among the land uses.

6) End of Introduction.  Since this journal style runs Intro --> Results, I think you need a paragraph at the end of the Intro which sets out what you will do, and how your information will be structured.  For example

"In this study, in order to test xyz, we measured core physiological (%C, %N) and isotopic (13C, 15N) parameters of buffel grass tussocks sampled from 30 sites in the north west of Mexico" or something like this.

Thank you for this suggestion. It is awkward, indeed, to jump from the introduction to the results without an idea of what was done. We followed the recommendation and have amended the introduction accordingly.

Minor

Sometimes the writing is a little slack.  For example, the first sentence of the Introduction tends to run-on, and should be split into two sentences. 

We've reworded the first sentence of the introduction and split it into two separate ideas.

"Extensions of territory" (ln40) could be simplified to "area".  (indeed, I feel this whole sentence is backwards in structure).

Thank you for this suggestion, which we have implemented in the revised manuscript.

Ln57 - "In this respect" -- I'm not sure that you mean. Perhaps you mean "To address this limitation..." or similar?

Thank you for this suggestion, which we have included in the revised manuscript.

Ln 71 - "and their intensive and polluting activity" - this feels out of place.  It also feels like something which needs some evidence regarding local conditions.

Indeed, this wording invoked the same non-existing information that we were unable to provide in Major Point no. 3 above. We have reworded this sentence citing the environmental regulation that requires cities of certain sizes to have monitoring networks.

Ln114: "28.8 f"

Checked and corrected.

Ln120;  "were as low as -15.9‰"  --> "ranged from -15.9‰ ... .... to -13.7‰"

We rewrote the phrase for clarity following the reviewer's suggestion.

Ln127: "cultivation fields" --> cultivated fields

Checked and corrected.

 The discussion was rather well written, but I felt it was a little more hypothetical, rather than dealing with the phenomena in your data.  I take issue with a couple of points.

We appreciate this Reviewer's assessment of the writing. We have modified the discussion to be more factual and have addressed the five points as follows:

1) The fact that Buffel grass is C4 should have been brought up in the intro, not the discussion

Thank you for pointing this out. We have indicated in the revised Introduction that buffelgrass utilizes C4 photosynthesis.

2) Ln150/1 -- discussing isotopic discrimination with respect to aridity "which was not oberved here" -- you never show us this in the data -- it's just an assertion.

Following major point No. 2, we can now claim that there were no underlying climatological/soil differences among the various land-use types. However, we have reworded this sentence because it was unclear whether we did not find differences in precipitation or in isotopic discrimination.

3) Ln215 - you talk about soil saturation.  Did you measure it?  Where was that reported in the data?  I didn't see it.

The reviewer is correct, we did not measure the soil saturation, instead we use the 15N enrichment as a proxy for chronic nitrogen accumulation, which can lead to saturation. This is based on the fact that the more nitrogen available in the soil, the more enriched it  becomes. This has been discussed in better detail in the revised discussion.

4) Ln220 -- "the likely result" -- how did you determine this was the most likely result?  I could not do that based on what you showed us!

Indeed, the wording was confusing: we didn't mean that this was the most probable result, but that our observations probably responded to fractionation. We have rewritten the paragraph for clarity and strived to remove confusing wording. 

5) Ln226 -- "threshold of saturation" -- my review notes just say "prove it".  I don't see any evidence of this in your manuscript. 

Indeed, we didn't measure the availability of nitrogen nor the nitrate reductase activity, so this claim was really out of order. We have reworded this paragraph of the revised discussion and have indicated that further research is required that measures the rate of deposition and the nitrogen tolerance of our study species. Admittedly, the wording of the original manuscript appeared to claim various unsubstantiated issues. This has been corrected in the revised manuscript, where we have been careful to distinguish what is known from the literature and how it may have influenced our results.

Overall, I think you need to present more of your data, so that readers can understand and draw their own conclusions.  At the moment, it feels like you're just telling us a story, and we are simply required to believe you.

This observation is consistent with the initial request of more data transparency. We hope that the revised analyses, additional figure, and supplementary information, now give a clear idea of what we found.

I apologise if my comments seem strong.  However, I do think you have interesting data, but I am currently unable to put your numbers into any context, and it's very frustrating.

No apologies needed. When read in sequence, one can actually perceive the increasing frustration of the reviewer in their comments. Hopefully we have responded in a satisfactory manner and that they find the revised manuscript to be clearer and more factual.

Round 2

Reviewer 1 Report

General comment

The authors have done a great job when revising their manuscript. My points of criticism on an earlier manuscript version have been addressed in a satisfactory manner.

Author Response

Thank you for your comments on the previous version that helped us to produce a clearer manuscript.

Reviewer 2 Report

Dear authors,

This paper has been hugely strengthened by the author's hard work in re-writing.  I think with a few minor suggestions, it can be accepted for publication.

Introduction,

I suggest the authors insert a paragraph introducing the use and theory of isotope measurement in biomonitoring.  The end of the second Discussion paragraph explains %C / 13C well, but should probably be in the intro.  Later in the discussion, you explain a bit about the factors affecting %N / 15N, which I feel would be more effective if you introduce the underlying theory in the Intro.

second paragraph

"gold standard" may be more common than golden standard

fourth paragraph

reasons for which it was introduced  (please add the letter "s")

Results

HUGE improvement.  Use of PCA analysis is very good. I would suggest two things

1) put all your mean values for %C, 13C, %N, 15N in a supplemental table

2) where you say (for example) there was a land use effect, it would be more informative to say something more like "delta 15N levels were significantly more positive in medium sized urban areas than in the other environments". Rather than listing off data values (leave that for table S1), I feel it would be better to highlight trends.  You've done this very well throughout, but there are still a few which could be improved.

I very much like Fig 4.

Discussion

I felt the first paragraph was a bit weak - mainly covering some details which might fit better elsewhere.  The sentence starting "Invasive species..." should probably start "Buffelgrass..."

Paragraph 2

"However, considering that" --> "Given that"

"can be attributed" --> "were assumed to be..." (Maybe?)

Last sentence of Paragraph 3 -- suggest you insert "increased" before "nitrogen availability"

Paragraph 4

sentence 2; delete "For example," and start the sentence "Plants growing..."

In this paragrpah, I think "concentration" probably is more technically correct than content.

Delete "with nitrogen deposition" from the end of the final sentence.  seems repetitive.

Paragraph 5

When discussing lightning and biological fixation by legimes, I suggest 
"whose isotopic values are close to zero" --> "which do not discriminate between isotopes"

In the following sentence "main role" should be changed to "large role" or "have a large influence" or similar

Paragraph 6

"these compounds are 15N depleted" --> these compounds tend to be 15N depleted  -- I think a little safer language

I don't understand that sentence about Mexico city -- could you rephrase it please?

A couple of times you use "Buffel grass" - which should probably be "buffelgrass" (no space: just for consistency)

Paragraph 7

I feel this doesn't advance your narrative much, but may be more effective with an "isotopes" paragraph in the intro, allowing a bit of refocusing.

Para 8

"Both N2O and NH3 are gasses that can be" --> Both N2O and NH3 can be

"agricultural sites can have multiple origins" - can --> may

"do not have a significant effect" --> have been shown not to have a  significant effect (slightly stronger phrasing, emphasizing the evidential basis of your claim)

Para 9

"because a negative feedback can reduce of even halt further assimilation of nitrogen beyond a physiological threshold that is species specific and had been documented" -->  "because negative feedback mechanisms can reduce or even halt further nitrogen assimilation beyond species-specific physiological thresholds, which have been documented..."

P10

Strong final paragraph!

Big improvement!  I suggest you think about my comments and consider which you may wish to implement.  Happily, this read through was not frustrating at all.

Author Response

Reviewer 2 Comments and Suggestions for Authors Dear authors, This paper has been hugely strengthened by the author's hard work in re-writing. I think with a few minor suggestions, it can be accepted for publication. Thank you for your suggestions on this and the previous version. We feel that the resulting manuscript is clearer. Please see our responses to your comments on this version below. Introduction, I suggest the authors insert a paragraph introducing the use and theory of isotope measurement in biomonitoring. The end of the second Discussion paragraph explains %C / 13C well, but should probably be in the intro. Later in the discussion, you explain a bit about the factors affecting %N / 15N, which I feel would be more effective if you introduce the underlying theory in the Intro. You are right, an introduction on stable isotopes in biomonitoring is in order. We have expanded on this idea in the revised introduction. Most introductory material from the previous Discussion was moved to the end of the second paragraph δ13C and δ15N. However, we kept the second paragraph from the Discussion, because it was something that the other Reviewer requested in the previous round. second paragraph "gold standard" may be more common than golden standard Indeed, thank you for pointing this out. We have changed the wording. fourth paragraph reasons for which it was introduced (please add the letter "s") We've made the reasons plural. Results HUGE improvement. Use of PCA analysis is very good. I would suggest two things Thank you for suggesting an environmental characterization. We are also satisfied with the PCA. 1) put all your mean values for %C, 13C, %N, 15N in a supplemental table We have produced this data table and mentioned in figure 4 as supplementary material. 2) where you say (for example) there was a land use effect, it would be more informative to say something more like "delta 15N levels were significantly more positive in medium sized urban areas than in the other environments". Rather than listing off data values (leave that for table S1), I feel it would be better to highlight trends. You've done this very well throughout, but there are still a few which could be improved. We followed your advice on this issue and have included short descriptions of the trends. I very much like Fig 4. Thank you, it was created following your suggestions to the previous version. We agree that it allows to actually know at a glance what we found in the study. Discussion I felt the first paragraph was a bit weak - mainly covering some details which might fit better elsewhere. We have edited this paragraph and appended the "orphan" text from the isotopes paragraph that got moved to the introduction. We agree that the carbon content responses were not as is interesting as those for the other parameters, but we felt that we needed to keep this paragraph to maintain the sequence from the M&M's and the Results sections. The sentence starting "Invasive species..." should probably start "Buffelgrass..." Yes, this sentence works better when focused on the grass. Paragraph 2 "However, considering that" --> "Given that" "can be attributed" --> "were assumed to be..." (Maybe?) ok. We have reworded the sentence. Last sentence of Paragraph 3 -- suggest you insert "increased" before "nitrogen availability" Thanks. It does work better. Paragraph 4 sentence 2; delete "For example," and start the sentence "Plants growing..." ok. We've made this change. In this paragraph, I think "concentration" probably is more technically correct than content. Indeed, we have changed "content" to "concentration" six times in this paragraph. And reworded the buffelgrass sentence at mid paragraph which awkwardly utilized the "nitrogen content" phrase three times. Delete "with nitrogen deposition" from the end of the final sentence. seems repetitive. Indeed. We've reworded this final sentence. Paragraph 5 When discussing lightning and biological fixation by legumes, I suggest "whose isotopic values are close to zero" --> "which do not discriminate between isotopes" Ok. We've changed the phrase in the revised manuscript. In the following sentence "main role" should be changed to "large role" or "have a large influence" or similar Ok. We used "large role". Paragraph 6 "these compounds are 15N depleted" --> these compounds tend to be 15N depleted -- I think a little safer language indeed. We've made this change in wording. I don't understand that sentence about Mexico city -- could you rephrase it please? These should have been two sentences actually. We have corrected the punctuation, rephrased, and deleted the mention to Mexico City (which was intended to show an example of a city with "small" amounts of reduced N species, but it was distracting). A couple of times you use "Buffel grass" - which should probably be "buffelgrass" (no space: just for consistency) Thank you for pointing these out. We thought we had caught these. We searched and replaced four instances of "buffel grass" to "buffelgrass". Paragraph 7 I feel this doesn't advance your narrative much, but may be more effective with an "isotopes" paragraph in the intro, allowing a bit of refocusing. As mentioned above, this paragraph was actually requested by the other Reviewer, so we kept it. Para 8 "Both N2O and NH3 are gasses that can be" --> Both N2O and NH3 can be "agricultural sites can have multiple origins" - can --> may "do not have a significant effect" --> have been shown not to have a significant effect (slightly stronger phrasing, emphasizing the evidential basis of your claim) Para 9 "because a negative feedback can reduce of even halt further assimilation of nitrogen beyond a physiological threshold that is species specific and had been documented" --> "because negative feedback mechanisms can reduce or even halt further nitrogen assimilation beyond species-specific physiological thresholds, which have been documented..." ok. Thank you for these suggestions which we have implemented. P10 Strong final paragraph! Thank you. Big improvement! I suggest you think about my comments and consider which you may wish to implement. Happily, this read through was not frustrating at all. ok
